# Genetic variability of the endangered fish lake minnow (*Eupallasella percnurus*) in populations newly established by translocation and those existing long term in Poland

Jacek Wolnicki[1], Dariusz Kaczmarczyk[1]*, Justyna Sikorska[1], Rafał Kamiński[1], Adriana Osińska[2], Natalia Zawrotna[3]

1 Pond Fishery Department, National Inland Fisheries Research Institute, Poland, 2 Norwegian University of Life Sciences, Norway, 3 Medical University of Gdańsk, Gdańsk, Poland

☉ These authors contributed equally to this work.

* d.kaczmarczyk@infish.com.pl

**Data Availability Statement:** The data is held in a public repository include genotyping data submitted as a pdf file: Lake-

## Abstract

The lake minnow *Eupallasella percnurus* is a small leuciscid fish. In Poland, this species has been in a continuous decline since the mid-20th century and is presently considered as a extremely endangered. According to Polish law, *E. percnurus* is a strictly protected species that requires active conservation measures. In Poland, one the most common and effective measure of active protection *E. percnurus* is initiation of new populations. For this purpose, in 2004–2012, juvenile individuals originating from aquaculture conditions were translocated to group of isolated water bodies not inhabited by this species. The juveniles were offspring of parental fish belonging to the same local population, which is extinct at present. Five of those attempts were successful. The aim of the present study was to assess the genetic variation in a group new populations and compare genetic variation indicators with 13 old populations that had existed for decades. The polymorphism of 13 microsatellite markers was investigated, significance of differences in the genetic variation indicators between the groups were tested using a one-way analysis of variance (ANOVA). The mean values of all summary statistics under study, i.e. observed heterozygosity, expected heterozygosity and the total number of alleles, were higher in the group of new populations compared to almost all old ones. A similar dependence was found for Garza—Williamson *M* values, where the mean for the group of new populations was higher than in almost all old populations. Our results indicate that all recently established *E. percnurus* populations have not yet experienced any extensive founder effects or bottlenecks. They have preserved a large part of the genetic variability typical of their maternal population, which might also have been relatively high. This feature of new populations, may give them a relatively high ability to adapt to changing environments in the future.

minnow_genotyping_MSA_ready.pdf This file is available at https://osf.io/62tby/?view_only= 513c41b6d7e547c0a558b116a0a8374e.

**Funding:** 1. Ministry of Science and Higher Education, Poland, Grant Number: N N304 324839 for 2010–2013; (financial support of: collection of samples, lab analysis, chemicals, NGS sequencing), 2. National Science Centre, Poland, Grant Number: 2014/15/B/NZ9/05240 for 2015–2019; (financial support of: collection of samples, lab equipment, lab analysis, lab plastics, chemicals) 3. Statutory Research Topics of the National Inland Fisheries Research Institute, Poland, Grants Numbers: Z-005 and Z-020 for 2024-2026. (financial support of: prepare the manuscript, publication costs).

**Competing interests:** The authors have declared that no competing interests exist.

## Introduction

Lake minnow *Eupallasella percnurus* (Pallas, 1814) is a small freshwater fish, a member of the Leuciscidae family, widely distributed in the Northern Hemisphere. The range of this species extends from the Oder River basin in Western Poland, throughout the northern part of Eurasia, to the Pacific coast and the Sakhalin and Hokkaido islands [1]. It is commonly accepted that this species is not at risk of extinction on a global scale [2]. In contrast, in Poland, *E. percnurus* belongs to the rarest and most imperiled freshwater fish species [3]. The major cause of this is the specific nature of its habitats. They are small (often 0.05–0.1 ha) and very shallow (0.5–1.0 m maximum depth) water bodies of anthropogenic or natural origin. The former constitute at least 70% of all water bodies and came into existence due to peat exploitation in the 20th century; the latter are frequently tiny pools in land depressions. All such water bodies have a limited period of existence—usually several decades—due to progressive shallowing resulting from intensive plant succession. The processes of vanishing habitats can be considerably accelerated by natural factors, such as droughts and heatwaves, and different forms of human activities, such as intentional draining, filling the water bodies with different materials and littering. As a consequence, a consistent decline in the number of *E. percnurus* sites in Poland has been observed since the mid-20th century [4]. Of the 100 sites described in the previous century, only around 10 survived until the onset of the present one [5]. Currently, *E. percnurus* populations occur at 110–120 sites in Poland [6], whereas a decade earlier their total number was assessed at 160–170 [3]. Due to the continuous disappearance of *E. percnurus* sites in the country, this fish has been strictly protected by law since 1975; in 2004, it became a species that formally requires active protection measures [7].

The active protection of *E. percnurus* commenced in 2002 within the framework of a long-term local project carried out in Central Poland, where this species was almost absent [8]. The main goal of the project was to establish several new *E. percnurus* populations by translocation of cultivated juvenile individuals into selected small water bodies not inhabited by this species. Fish juveniles were offspring of several tens of wild parental individuals, originating from a large local population inhabiting a vanishing mid-forest lake (52˚29’31.89" N; 21˚16’17.40" E). To obtain juvenile fish, standard propagation techniques were used [9, 10], and techniques for controlled rearing were developed through *E. percnurus* larval and juvenile stages [11, 12].

In 2004–2012, juvenile *E. percnurus* were released into several isolated water bodies to initiate new populations, and today five of these attempts can be considered entirely successful [8]. The annual monitoring of the new populations demonstrated that they required only 3–4 years after the final or the only translocation to stabilize both the population size and its sex structure with a strong predominance of females over males [13, 14]. However, to comprehensively evaluate the prospects of new *E. percnurus* populations for survival in the future, knowledge of their genetic variability is indispensable. It is especially important to consider that the vast majority of the Polish populations of this species, of an adequately long period of existence (several tens of years), might have experienced founder effects and/or strong genetic bottlenecks, resulting in a reduction in their genetic variation. In such cases, their ability to adapt to the changing environment [7] can be compromised.

Therefore, in the present work, we intended to comparing genetic diversity of old and newly established populations in order to evaluate from a conservation genetics perspective the ongoing translocation programme. Moreover, we tried to determining the genetic diversity of natural populations to identify potential sources for future captive-breeding programmes for translocation or reinforcement.

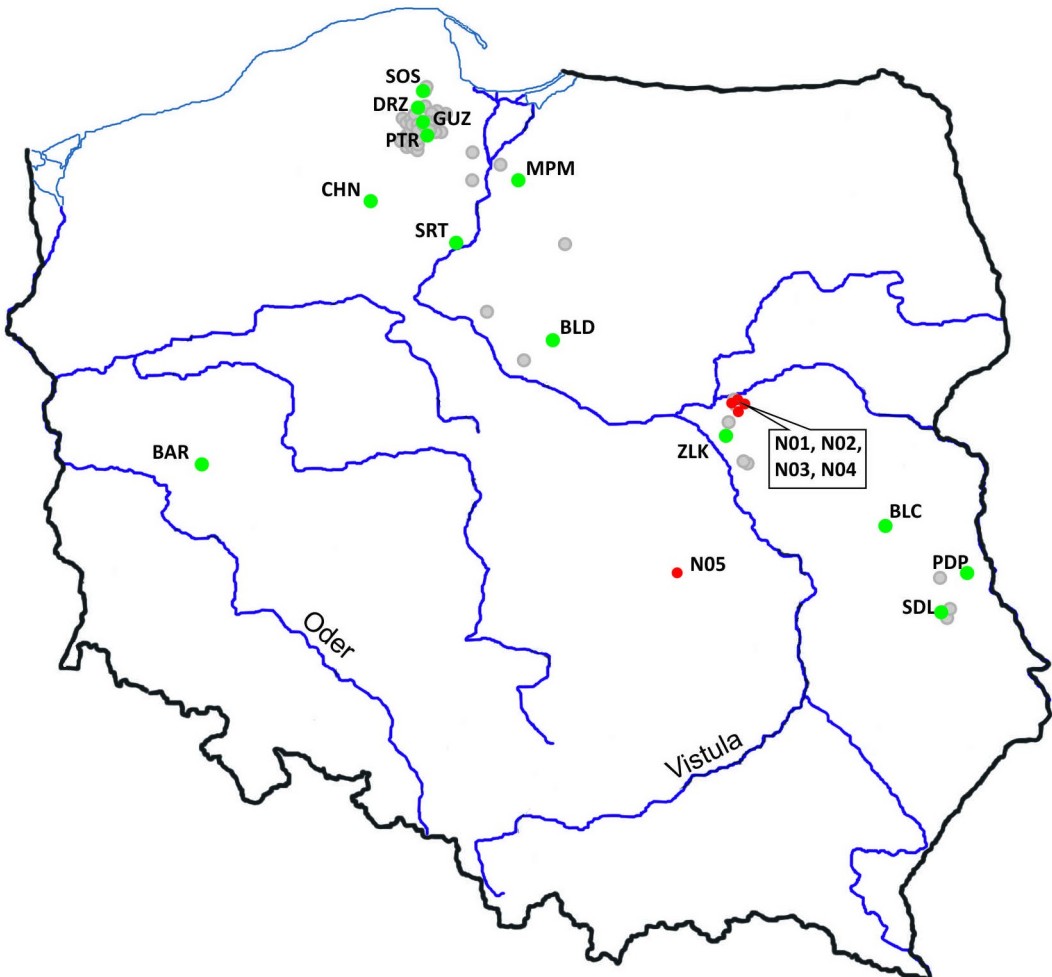

**Fig 1. Present distribution of *E. percnurus* sites in Poland (grey circles).** Red circles indicate new populations (N01-N05) established by fish translocation; green circles indicate investigated populations existing over decades: Barłożnia (BAR), Bełcząc (BLC), Bledzewo (BLD), Chojnice (CHN), Drozdowo (DRZ), Guzy (GUZ), Mikołajki Pomorskie (MPM), Piotrowo (PTR), Podpakule (PDP), Sartowice (SRT), Siedliszcze (SDL), Sośniak (SOS), Zielonka (ZLK). The nameless five water bodies where new *E. percnurus* populations were initiated differed considerably in size and water reaction (pH) values but not in water depth (Table 1).

## Material and methods

### Study area, translocations and fish sampling

Genetic studies comprised a total of 18 *E. percnurus* populations from throughout its entire range of occurrence in Poland; among them, five were established recently, and 13 came into existence in the 20[th] century (Fig 1). The minimum age of the old populations was assessed at 30–40 years, with the maximum probably exceeding 50 years. Some of the old populations were investigated during our previous studies [7].

Translocations with the use of 0+ aged juveniles were performed once or repeated in consecutive years. The maternal population was situated 2–4 km away from new populations N01-N04, and about 120 km from N05. The total number of juveniles released to the individual water bodies varied between 470 and 3,300. Samples of fish from new populations for the genetic studies were taken 4–8 years after the final or the only translocation. Fish from all

**Table 1. Basic data on *Eupallasella percnurus* habitats, translocations and sampling.**

| Code name of a water body and new population | Latitude/ Longitude | Maximum water surface (ha) | Maximum water depth (m) | Water reaction (pH) | Number of fish translocated | Year of translocation | Year of sampling |
|---|---|---|---|---|---|---|---|
| N01 | 52°28'46.27"/21° 14'51.88" | 1.2 | 1.0 | 5.1–7.0 | 470 | 2012 | 2016 |
| N02 | 52°30'11.50"/21° 19'02.75" | 0.2 | 1.2 | 5.5–6.5 | 1.000 | 2009 | 2016 |
| N03 | 52°30'28.20"/21° 15'19.11" | 0.2 | 0.8 | 5.5–7.0 | 1.530 | 2004–2006 | 2012 |
| N04 | 52°29'38.75"/21° 19'45.45" | 0.02 | 1.0 | 7.0–8.0 | 400 | 2009 | 2017 |
| N05 | 51°30'01.71"/20° 40'50.41" | 0.5 | 1.2 | 6.5–7.5 | 3.300 | 2007–2008 | 2012 |

investigated populations were captured using baited traps with two openings (25 × 25 × 40 cm; mesh 5 mm; opening diameter 60 mm) [13]. The access to the particular water bodies in the field was based on the verbal permission of the land owners. To maintain fish welfare and minimize their stress, all manipulations with the captured fish were preceded by their anesthetization in a water solution of MS-222 (Argent Labs, USA) at a concentration of 80 mg/dm$^{-3}$, and shortened to the minimum (15 min). 48 adult fish from each population were sampled. The biological materials for genetic analyses were little (3–5 mm; 10–20 mm$^2$) tips of the left pelvic fins. The fish were awakened from anesthesia in fresh water and released immediately into their home water body. All procedures of fish catching and sampling were approved by the Local Ethics Commission in Olsztyn, Poland (Decision No. 22/2010 of January 27, 2010 for 2010–2012 and Decision No. 15/2015 of March 25, 2015 for 2015–2018). The wet fin fragments were put on a stiff 15 × 12 cm plate wrapped in aluminium foil and allowed to air dry. In a laboratory, the samples were placed in an air- dryer to remove the humidity at 30°C. Dry fin fragments were wrapped in a new piece of aluminium foil, numbered and stored separately in 1.5 ml Eppendorf tubes.

## DNA extraction and microsatellite amplification

Genomic DNA was extracted from fin tissues using the Genomic Mini AX Tissue SPIN DNA Extraction and Purification Kit (A&A Biotechnology, Poland). The extraction procedure followed the manufacturer's recommendations. The assessment of genetic variation was based on 13 polymorphic microsatellites: *Z9878*, *10362*, *13419* (primer sequences were taken from the National Center for Biotechnology Information (NCBI; http://www.ncbi.nlm.nih.gov), *Ca3*, *4* and *12*, [15, 16] and *Eupe1*, *2*, *4*, *5*, *6*, *7* and *9* developed by Kaczmarczyk and Gadomski and included in [17]. The forward primer of each primer pair was 5' end labelled with fluorescent dyes (6FAM, VIC, NED and PET). The composition of the PCR mixture and the thermal profiles of the PCR reaction were followed by [7]. The length of the amplified DNA fragments was determined using an Applied Biosystems 3130 Genetic Analyser against the GS500LIZ size standard. Fragment size and allele determination were performed using GeneMapper 4.0 software (Applied Biosystems). The number of genotyped samples was 48 for each population and was the same across all populations.

## Analyses of genetic diversity

In each population, genetic diversity was evaluated based on the observed ($H_o$) and expected heterozygosity ($H_e$) at a given locus and across loci. A mean number of alleles across loci ($AN$)

was determined. All calculations were computed using MSA software [18]. The Exact Hardy—Weinberg (H-W) test [19] was used to test the deviation from the H-W equilibrium. These calculations were performed using Arlequin 3.2.5 software [20, 21]. The likelihood of the null alleles presence at investigated loci was checked by using Hardy-Weinberg test for heterozygote deficiency with the Monte Carlo randomization [19]. The probability was assessed and the 'U' test was performed as described by [22]. A number of iterations in this test was set to 100000. Probability of null alleles presence was evaluated as significant if $p < 0.01$. The monomorphic loci as well as loci almost monomorphic with very low frequency ($f < 0.035$) of allele different than main allele were excluded from those calculations. The tetrasomic locus *CA4* was also excluded from this test. Those calculations was performed by using ML-Null Freq v.1.0 software [23]. Detection of loci under selection was based on the F-statistic, distance method for AMOVA computation and number of different alleles ($F_{ST}$ like). Number of simulations was set on 50000 and number of demes 18. Significance level of this test was set at $p \leq 0.05$. These calculations were performed using Arlequin 3.2.5 software [20, 21]. The likely occurrence of a bottleneck or founder effect and their influence on within-population genetic variability were based on the Garza—Williamson $M$ index [24], including Excoffier's adjustment [20, 21].

## Analyses of genetic structure

Identification of factors that may contribute to genetic variations in groups of new and old populations was attempted via Bayesian analysis of population structure with STRUCTURE 2.3.4 software [25]. This analysis focused on detecting the possibility of admixtures between populations, and on assigning fish to their source population or other populations. To test the K parameter, it was increased stepwise from 8 to 20 (18 populations plus 2 suspected subpopulations). The length of the Burn-in Period was set at 100,000, and the number of Markov Chain Monte Carlo (MCMC) reps after Burn-in was also 100,000. Each of those simulations were repeated four times. The InPD values were calculated as a average of all four iterations of STRUCTURE runs at given K. The optimum K value used in the interpretation of the STRUCTURE results was calculated according to the Evanno method [26] using Structure Harvester Tool [27], available at https://taylor0.biology.ucla.edu/structureHarvester/ (accessed at 01.01.2024).

In an analysis of differences between both groups of populations we also performed a comparison of their allelic diversity, calculation of the overall number of alleles in the groups, and identification of private alleles specific for given population or group of populations. Those calculations were performed by using using GenAlEx v.6.5 [28].

## Data analyses

The means of indicators of genetic variation ($H_o$, $H_e$, $AN$) and $M$ value of the Garza—Williamson index were compared between the group of new populations and the older group of populations. Before starting a statistical analysis of differences between new and old *E. percnurus* populations, the distribution of means was tested using the Shapiro—Wilk test at the significance level $\alpha = 0.05$. Those tests confirmed the normal distribution of all variables, so, for this reason, we used the parametric test analysis of variance (one-way ANOVA). These calculations were performed using STATISTICA 13.3 software (StatSoft Inc., USA). The number of Degrees of Freedom was 15 for the former and 17 for the latter variant. The significance threshold for the differences between the groups was set as $p \leq 0.05$; differences were identified as highly significant at $p \leq 0.01$.

**Table 2. The values of genetic variation indicators and Garza-Williamson index in new populations established by fish translocations (N01-N05) and old populations existing over decades (abbreviations in alphabetical order).**

| Population code | Observed heterozygosity ($H_o$) | Expected heterozygosity ($H_e$) | Total number of alleles ($AN$) | Garza-Williamson index ($M$) | Source |
|---|---|---|---|---|---|
| N02 | 0.53 | 0.49 | 39 | 0.54 | this work |
| N01 | 0.49 | 0.45 | 40 | 0.58 | this work |
| N03 | 0.56 | 0.50 | 45 | 0.65 | this work |
| N04 | 0.40 | 0.41 | 35 | 0.46 | this work |
| N05 | 0.45 | 0.46 | 39 | 0.43 | this work |
| mean ± SD (SE) | 0.49 ± 0.06 (0.03) | 0.46 ± 0.04 (0.02) | 40 ± 3.6 (1.60) | 0.53 ± 0.09 (0.04) | |
| BAR | 0.38 | 0.35 | 31 | 0.44 | [7] |
| BLC | 0.47 | 0.47 | 53 | 0.48 | [7] |
| BLD | 0.33 | 0.28 | 34 | 0.57 | [7] |
| CHN | 0.25 | 0.27 | 30 | 0.40 | [7] |
| DRZ | 0.33 | 0.31 | 27 | 0.37 | K & W (unpubl.) |
| GUZ | 0.39 | 0.34 | 33 | 0.35 | K & W (unpubl.) |
| MPM | 0.16 | 0.15 | 23 | 0.38 | K & W (unpubl.) |
| PDP | 0.60 | 0.55 | 44 | 0.58 | [7] |
| PTR | 0.28 | 0.26 | 30 | 0.33 | [7] |
| SDL | 0.23 | 0.20 | 22 | 0.24 | K & W (unpubl.) |
| SOS | 0.41 | 0.25 | 34 | 0.53 | [7] |
| SRT | 0.34 | 0.30 | 36 | 0.43 | [7] |
| ZLK | 0.32 | 0.31 | 28 | 0.44 | [7] |
| mean ± SD (SE) | 0.35 ± 0.11 (0.03) | 0.32 ± 0.10 (0.03) | 33 ± 8.3 (2.31) | 0.43 ± 0.10 (0.28) | |
| w/o BLC and PDP mean ± SD (SE) | 0.31 ± 0.08 (0.02) | 0.28 ± 0.06 (0.02) | 30 ± 4.5 (1.40) | 0.41 ± 0.09 (0.27) | |

K & W (unpubl.)–Kaczmarczyk D., & Wolnicki J. unpublished data

## Results

### Analyses of genetic diversity

Among the new populations, the individual values of $H_o$ and $H_e$ were similar and ranged from 0.40 (N04 population) to 0.56 (N03 population) (SD 0.07) and from 0.41 (N04 population) to 0.50 (N03 population) (SD 0.03), respectively (Table 2). In the group of old populations, the individual values of these indicators were generally lower and differed considerably, ranging from 0.16 to 0.60 (SD 0.12) for $H_o$ and from 0.15 to 0.55 (SD 0.11) for $H_e$. In two old populations (BLC and PDP), the values of $H_o$ and $H_e$ were relatively high and close to the data recorded for the new populations. For new and old groups of populations, the mean values of $H_o$ were: (0.49 v. 0.35, respectively) and $H_e$ were (0.46 v. 0.32, respectively). The $H_o$ values differed significantly (F-Stat = 6.8756, p = 0.0185) between both groups, but after excluding BLC and PDP populations, those differences were highly significant (F-Stat = 20.3017, p = 0.0005). The differences in the mean $H_e$ values between the groups of new and old populations were highly significant (F-Stat = 8.8549, p = 0.0089), and after excluding populations BLC and PDP, their significance increased (F-Stat = 34.5104, p < 0.0001). In all populations, the mean $H_o$ values were close to the mean $H_e$ values under H-W equilibrium. When calculated across all

markers, departures from this equilibrium were not significant ($p > 0.05$). Departures were found only at the level of individual loci. Departures at locus *Z9878* were significant in most populations, but significant deviations at other loci were detected only in some populations. It is likely that departures from H-W equilibrium at locus *Z9878* are the consequence of the selection process ($F_{ST}$ $p < 0.05$) of a gene located near to this microsatellite and thus closely related with its alleles. Moreover, significant traces of selection processes that can affect frequency of alleles were found in three other loci: *CA3*, *Eupe1* and *Eupe9* ($F_{ST}$ $p < 0.05$). In the case of other loci no significant traces of selection processes were observed. Investigated loci differed between populations in probability presence of null alleles. A significant probability of null alleles presence ($p < 0.05$) was observed at *Z10362* (BLC, and SRT), *Z13419* (BAR), *CA3* (LOJ), *CA12* (CHN), *Eupe1* (N04), *Eupe2* (N05), *Eupe4* (CHN), *Eupe6* (N04, N05), and *Eupe7* (N05). In other populations the probability of null alleles was not significant, therefore a potential existence of null alleles at investigated loci seems to have only an limited effect to genetic characteristics of investigated populations.

On the level of individual populations belonging to the new group, the total number of alleles (*AN*) detected across all loci ranged from 35 (N04 population) to 45 (N03 population) (SD 3.6), whereas 22–53 (SD 8.3) alleles were recorded in individual populations that belong to the group of the old group (Table 2). The difference between the respective mean number of alleles for the groups (40 *v.* 33) was not significant (F-Stat = 3.1118, p = 0.0968); however, after excluding populations BLC and PDP, which had the highest *AN* value (53 and 44, respectively), they were highly significant (F-Stat = 18.0696, p = 0.0008).

A genetic variation in all investigated populations was reduced as a result of bottlenecking or founder event (Garza—Williamson *M* index < 0.7), but the sizes of the reductions differed. The individual values of the Garza—Williamson *M* index proved to be higher in new populations (range 0.43 (population N05)– 0.65 (population (N03); SD 0.09) compared to the old populations (0.24–0.58; SD 0.10). The difference in the mean *M* value for the group of new and old populations (0.53 *v.* 0.43; Table 2) was not significant (F-stat = 4.431, p = 0.0515), but it became significant (F-stat = 6.4827, p = 0.0233) when BLC and PDP were excluded.

## Analyses of genetic structure

The analysis performed using STRUCTURE 2.3.4. software (at K values from 8 to 17) revealed a similarity of genetic characteristics of fish in new populations N01 –N05. Differences between them and populations N05 started at K11 and N03 at K = 13. Admixture traces at K range (8–20) were recorded in populations BLC and PDP, but the former turned out to be the only one with distinct traces of admixtures at K values ranging from 13 to 20. In other populations, admixture traces were not clear. Some of the traces started at K = 16 or K = 17 and disappeared at higher K values. K = 11 was identified as the optimum value of this parameter because at this level the population structure, their relationship and admixture status were very close to field observations and results of our previous studies. On the other hand, the Delta K analysis performed by using Evanno method (Fig 2) indicated optimal K = 13, which shows own clusters for N03 and N05 populations. Over analyzed K values group of new populations remain in the same cluster up to K = 10. Starting from K = 11 the population N05 that had five private alleles (specific for this population and not observed in the others from this group) starts its own cluster. The population N03 had four private alleles and starting from K = 13 became different from other new populations. The other three new populations stay in the same cluster up to K = 17 when population N04 creates its own cluster. The analysis performed by GenAlEx software revealed that in new populations only five out of 59 alleles were private and not observed in old populations. Among them only one allele was detected in all new

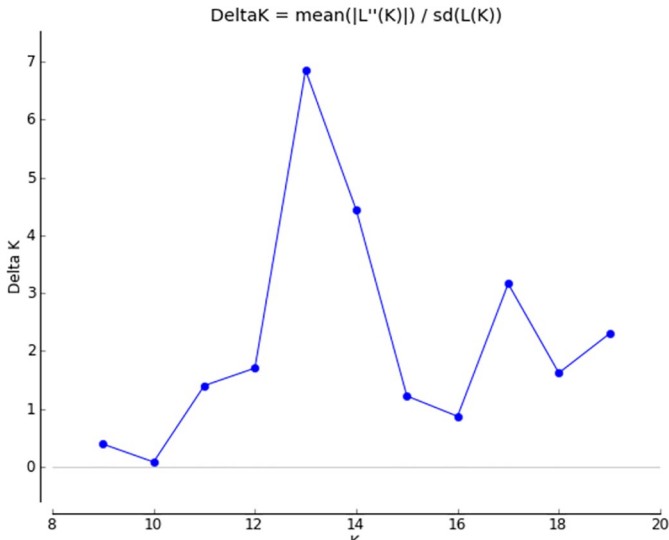

**Fig 2. Results of Delta K analysis performed by using Evanno method.**

populations, one allele was identified in three out of five populations, and other three alleles in one of the new populations. As reported above, only a few alleles were identified as population specific. Most of the alleles detected in this group were common alleles and were found in the majority of populations belonging to this group. Consequently, the populations inside this group differed mostly in the frequency of common alleles than in the presence of population specific alleles. Inside the old populations 93 alleles were identified, but 25 of them were private for this group. Moreover none of them was observed across the entire group of 13 populations that comprise the old group. Only one of them was detected in seven out of 13 populations. Most private alleles were population specific and detected in one population or in two, maximum three of them. Summarising, genetic differences inside this group came from the existence of numerous private alleles and differences in frequency of common alleles, resulting in a high genetic distance between them.

Those two bar plots as well as other showing the probability of assigning an individual fish to a given population at K 8, 10, 11, 13, 15, 17 were shown in Fig 3.

## Discussion

The origin and age of *E. percnurus* populations and habitats that presently exist in Poland are well documented only in very rare cases. It is generally accepted, however, that the vast majority appeared as a result of human activity connected with peat extraction in the middle part of the 20th century [3, 29]). This makes it possible to assess the age of most *E. percnurus* populations at close to 50 years and about 25 fish generations. It seems then very likely that most of them, occurring in highly variable habitats (i.e. small, extremely shallow, vanishing water bodies), had to experience strong founder and/or bottleneck effects in the past. These effects resulted in a decline in their effective size and an increase in the rate of inbreeding, eventually resulting in a gradual decrease in their genetic variability. This phenomenon was favored by factors such as the distinct geographical isolation of most of the populations, the non-migratory nature of the species and a general lack of translocation of the fish, excluding translocations over very short distances carried out by members of the local communities [7].

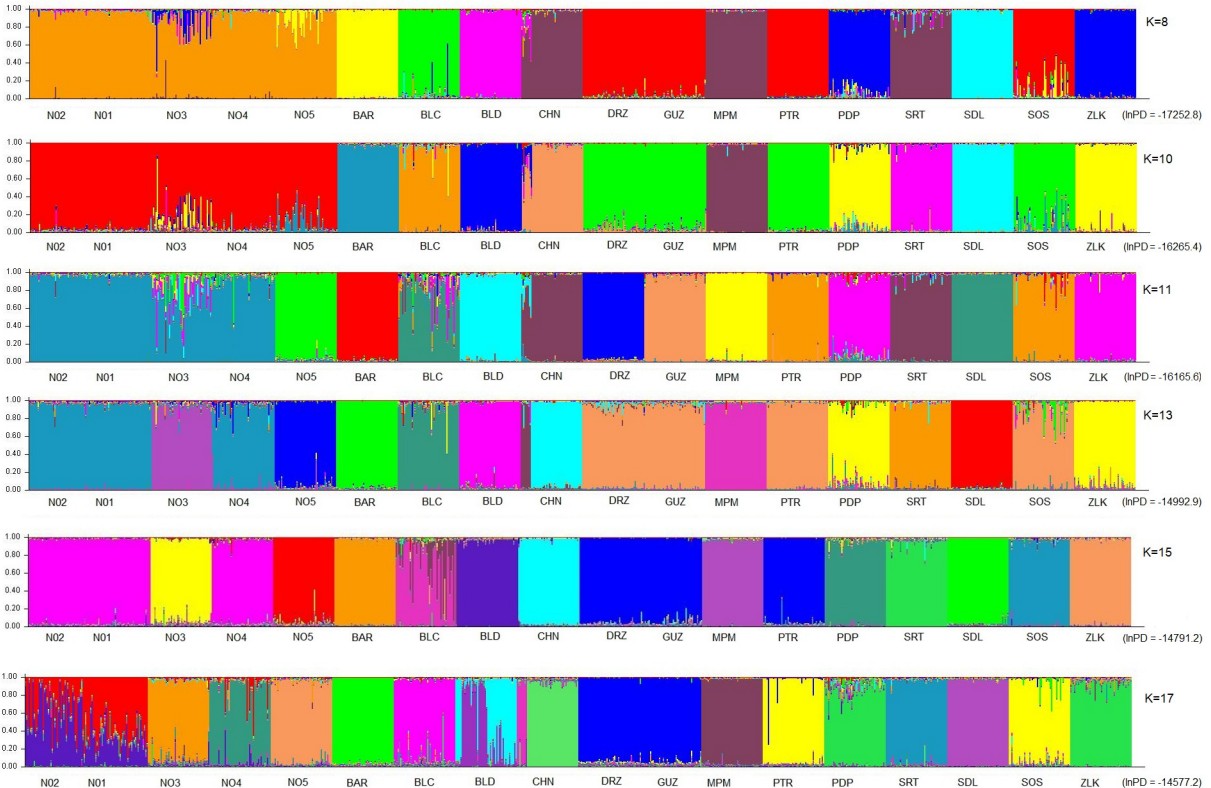

**Fig 3. Results of admixture analysis performed using STRUCTURE software.** New populations: N02, N01, N03, N04, N05; old populations: Barłożnia (BAR), Bełcząc (BLC), Bledzewo (BLD), Chojnice (CHN), Drozdowo (DRZ), Guzy (GUZ), Mikołajki Pomorskie (MPM), Piotrowo (PTR), Podpakule (PDP), Sartowice (SRT), Siedliszcze (SDL), Sośniak (SOS), Zielonka (ZL). Average lnPD values are given in the brackets.

The results of genetic studies performed at the onset of the present decade [3] and those obtained quite recently (present work) are in good agreement with the findings mentioned above. The 13 old *E. percnurus* populations included in the present study were characterized by relatively low genetic variation indicators: observed and expected heterozygosity and the total number of alleles. Moreover, they all had an *M* index value lower than 0.6, which is additional evidence that old populations experienced severe genetic bottlenecks that resulted in a reduction in their size with further negative genetic consequences [22].

In the group of old *E. percnurus* populations, all indicators of genetic variation differed considerably. Two populations, BLC and PDP, presented the highest values of heterozygosity, and the largest allelic diversity detected. Moreover, the traces of admixtures suggest and confirm the fact that both populations might be parts of larger metapopulations. The reason for this can be their incomplete isolation from other closely located populations of the species and subsequent gene flow. In fact, their habitats are parts of small complexes of several water bodies where populations of this fish existed a decade ago (BLC) or exist at present (PDP). Presumably, during spring flood events, gene flow can occur among local populations within their complexes. If genetic flow between them occurs quite often, the spatial isolation of the resistant patches of the metapopulation does not affect the consequences of genetic drift and reduces the rate of their genetic divergence to that typical for one large and stable population. Moreover, the genetic variability of each patch of the metapopulation declines more slowly than in an isolated population of a similar size. Consequently, the current level of genetic variation in

the former population can result from the founder effect rather than from population bottle-necks [30]. The status of a metapopulation may have a mitigating effect on declines in hetero-zygosity and allele number caused by genetic bottlenecks. If lake minnow populations are part of larger metapopulations, such situations should not be considered exceptional. The two dis-cussed here (BLC and PDP) are the only identified metapopulations among those investigated in this paper, and they had a disproportional impact on the results of the statistical analysis. Therefore, their particular status could explain some bias in the comparison of the genetic vari-ation indicators between the groups of new and old populations. This justifies why we tested the significance of differences in values of the genetic variation for the population groups in two variants that included all old populations or excluded BLC and PDP.

In contrast, the lowest heterozygosity and very low values of *AN* and *M*, recorded for the old MPM population, seemed to be caused by total isolation throughout its long existence. It should be stressed that it occurs in a highly isolated mid-forest water body in an area where no other *E. percnurus* populations are likely to have ever occurred. Therefore, in its history, the feasibility of any gene exchange with another population of this fish should be considered extremely low. The effect of such a situation is a clear tendency to worsen the genetic variation observed during the last decade. A comparison of the heterozygosity indicators determined in 2011 [7] and 2016 (present paper) showed that in the meantime, the values declined from 0.21 to 0.16 ($H_o$) and from 0.18 to 0.15 ($H_e$), at the same very low total number of alleles (23).

Surprisingly, all newly established *E. percnurus* populations were of relatively high heterozy-gosity and allelic diversity, higher than almost all old populations. However, the differences between the former and the latter were not always significant. They become highly significant only when two old populations (BLC and PDP) were excluded from the group of old popula-tions. These two populations were in fact metapopulations, rather than individual populations strongly isolated from others. This phenomenon seems to arise from three different reasons. The first is the relatively high genetic variation in the population that served as a source of maternal fish used to obtain the progeny for translocation. This assumption cannot be verified because this population unexpectedly became extinct together with its habitat before genetic investigations could have been carried out. However, a comparable level of genetic diversity recorded in all new populations and in two old populations, BLC and PDP, with the highest genetic variation, can suggest the existence of high genetic diversity in the now extinct maternal population used for obtaining progeny for translocations. The second reason seems to be the short period of the existence of new populations prior to the genetic analyses in relatively stable habitats, which is known from monitoring. Hence, presumably, none of the new populations have experienced any serious genetic bottlenecks that could have strongly reduced their genetic diversity. The third reason may be the fact that to initiate new populations of this species, juve-niles at relatively high number were translocated. This made it possible to preserve a relatively large part of the genetic diversity of the maternal population in all-new populations. It is note-worthy, that N04 population was initiated by translocation of the lowest number of fish, and consequently it proved to have the lowest values of genetic variation indicators in this group.

Despite a progressive increases in genetic differences in the group of five new populations, they still maintain their general similarity. Most of the differences seem to result from random changes in allele frequency known as genetic drift. They seem to have a stronger effect than the existence population specific alleles. The increased rate of genetic divergence is especially noticeable between populations N03 and N05 where it overlaps with possible founder effect, thus resulting in a slightly different set of alleles detected in those populations. These differ-ences could have resulted from the use of different sets of spawners in each year as well as from their individual reproductive success. It should be stressed that N03 and N05 populations are the only ones where repeated translocations were used. Thus, two (N05) or three (N03) years

of translocations combined with relatively high number of fish used could have increased the genetic diversity as well as the differences between them and the remaining new populations. The differences between the latter seem to be caused by environmental factors that affect the genetic variation of each of them [31] and strong genetic drift that is typical for this species [7].

Although the genetic characteristics of fish used for translocations are unknown, a significantly higher $M$ value in the new populations versus old ones (without BLC and PDP) may suggest that both bottleneck effect in the source population and the founder effect inside new populations were relatively weak. This feature should positively affect their genetic viability in the future.

To the best of our current knowledge, all new lake minnow populations are strictly isolated from other populations of the species. Thus, gene flow between new and any other populations of this fish could not have been the factor contributing to their genetic variation. Consequently, the environmental factors and genetic drift are probably the key factors that determine the rate of differentiation inside new lake minnow populations and differences among them detected in these studies as well.

In the present work, the group of 13 old *E. percnurus* populations constitute approximately 11% of all its populations currently existing in Poland, so it can be regarded as a sample representative of the Polish genetic state of this species. This means that populations of this species of relatively high genetic variability that might be used for success in active protection programs are a considerable minority in the country. This makes knowledge of the genetic diversity of a potential maternal population decisive for the successful conservation of this species through translocations of its cultivated juveniles or wild individuals.

## Acknowledgments

We would like to thank G. Radtke (National Inland Fisheries Research Institute, Poland) for his valuable field assistance.

## Author Contributions

**Conceptualization:** Jacek Wolnicki, Dariusz Kaczmarczyk.

**Data curation:** Dariusz Kaczmarczyk.

**Funding acquisition:** Dariusz Kaczmarczyk.

**Investigation:** Dariusz Kaczmarczyk, Adriana Osińska, Natalia Zawrotna.

**Methodology:** Dariusz Kaczmarczyk.

**Project administration:** Jacek Wolnicki.

**Resources:** Jacek Wolnicki, Justyna Sikorska, Rafał Kamiński.

**Supervision:** Jacek Wolnicki.

**Writing – original draft:** Jacek Wolnicki, Dariusz Kaczmarczyk.

**Writing – review & editing:** Jacek Wolnicki, Dariusz Kaczmarczyk.

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
