## [Decision Letter · Decision Letter 0]

2 Feb 2024

PONE-D-24-01902Genetic variability of the endangered fish lake minnow (Eupallasella percnurus) in populations newly established by translocation and those existing long term in PolandPLOS ONE

Dear Dr. Kaczmarczyk,

Thank you for submitting your manuscript to PLOS ONE. After careful consideration, we feel that it has merit but does not fully meet PLOS ONE’s publication criteria as it currently stands. Therefore, we invite you to submit a revised version of the manuscript that addresses the points raised during the review process.

We look forward to receiving your revised manuscript.

Kind regards,

Tzen-Yuh Chiang

Academic Editor

PLOS ONE

Journal Requirements:

2. To comply with PLOS ONE submissions requirements, in your Methods section, please provide additional information regarding the experiments involving animals and ensure you have included details on (1) methods of sacrifice, (2) methods of anesthesia and/or analgesia, and (3) efforts to alleviate suffering

4. We suggest you thoroughly copyedit your manuscript for language usage, spelling, and grammar. If you do not know anyone who can help you do this, you may wish to consider employing a professional scientific editing service. 

"1.          Ministry of Science and Higher Education, Poland, Grant Number: N N304 324839 for 2010–2013;

2.            National Science Centre, Poland, Grant Number: 2014/15/B/NZ9/05240 for 2015–2019;

3.            Statutory Research Topics of the National Inland Fisheries Research Institute, Poland, Grants Numbers: Z-005 and Z-020 for 2024-2026. "          

7. When completing the data availability statement of the submission form, you indicated that you will make your data available on acceptance. We strongly recommend all authors decide on a data sharing plan before acceptance, as the process can be lengthy and hold up publication timelines. Please note that, though access restrictions are acceptable now, your entire data will need to be made freely accessible if your manuscript is accepted for publication. This policy applies to all data except where public deposition would breach compliance with the protocol approved by your research ethics board. If you are unable to adhere to our open data policy, please kindly revise your statement to explain your reasoning and we will seek the editor's input on an exemption. Please be assured that, once you have provided your new statement, the assessment of your exemption will not hold up the peer review process.

8. We note that Figure 1 in your submission contain map images which may be copyrighted. All PLOS content is published under the Creative Commons Attribution License (CC BY 4.0), which means that the manuscript, images, and Supporting Information files will be freely available online, and any third party is permitted to access, download, copy, distribute, and use these materials in any way, even commercially, with proper attribution. For these reasons, we cannot publish previously copyrighted maps or satellite images created using proprietary data, such as Google software (Google Maps, Street View, and Earth). For more information, see our copyright guidelines: http://journals.plos.org/plosone/s/licenses-and-copyright.

      a. You may seek permission from the original copyright holder of Figures 1 to publish the content specifically under the CC BY 4.0 license. 

Reviewers' comments:

Reviewer's Responses to Questions

**Comments to the Author**

1. Is the manuscript technically sound, and do the data support the conclusions?

Reviewer #1: Yes

Reviewer #2: Partly

2. Has the statistical analysis been performed appropriately and rigorously? 

Reviewer #1: Yes

Reviewer #2: No

3. Have the authors made all data underlying the findings in their manuscript fully available?

Reviewer #1: Yes

Reviewer #2: No

4. Is the manuscript presented in an intelligible fashion and written in standard English?

Reviewer #1: Yes

Reviewer #2: Yes

5. Review Comments to the Author

Reviewer #1: Line 122: Please revised Eupallasella percnurus to E. percnurus.

Line 248: Please change K 8-17 to K 8, 10, 11, 13, 15, 17. In line 240, why authors reported that K=11 was identified as the optimum value?

Reviewer #2: The manuscript contains an interesting study on the genetic diversity of the translocated populations of lake minnow Eupallasella percnurus in Poland. The topic is worthy of investigation, and the manuscript is particularly interesting in that it investigates the effect of conservation activities. In principle, the design of the study could be appropriate, but I have some questions that need to be clarified before publication.

The study aimed to investigate the conservation genetics of the populations resulting from the translocation program concerning the surviving original populations. Although the manuscript indicates that information on genetic diversity and structure of the original populations has already been published, this information should play a greater role in both the introduction and the evaluation. This is particularly true for the original population that is the source of the five new populations. There is practically no data available on this population in the manuscript, only its location and distance from the new populations, the number of individuals translocated, and the time of translocation are included. The strength and relevance of the manuscript to the broader interest is that translocations from known populations have been made with known numbers of individuals, and genetic analysis could show how successfully new populations can maintain diversity and the structure of the original population; at what initial size genetic drift, founder effects occur. However, it is not the other original populations that are needed to know this, but the genetic data of the source population.

The material and method are adequate for DNA extraction and microsatellite amplification. Still, the resulting database should allow for much more detailed analysis (unfortunately the raw data are not available in the submitted material, although the statement suggests that they will be available after publication.)

Before examining diversity and genetic structure, the presence of null alleles and loci under selection should have been investigated. The latter would also provide additional information for the objective.

In addition to calculating diversity indices, detecting private alleles and their frequencies in populations could provide important information. If data on the original source population are available, allele distributions can provide additional information on the changes in diversity. It would be important to know the diversity of the source population since each of the new populations mostly showed higher diversity than the 13 old populations.

Values of the G-W index less than 0.70 suggest that all of the new and old populations have experienced a bottlenecking or founder event.

Based on the genetic structure and diversity data, it appears that the translocations did not occur from a single source population in all cases. (The original lnPD values should be also presented for the Stucture analysis.) The delta K method indicated 13 clusters in the analysis, resulting in N03 and N05 being different from the other three new populations. Moreover, N03 shows the origin of the BLD population and N05 is a unique cluster, based on Figure 3. This latter one could be the result of private alleles remaining from the original population which are absent from the other three due to the founder effect; or the admixture with a local population that was not detected before the translocation. The results should be re-evaluated according to the findings from the Structure analysis.

The diversity data could be correlated with the translocated individual number (founder population size) to estimate its effect.

Although several issues need to be clarified to evaluate the results, in my opinion, once the details are provided and worked out, a valuable publication can be produced.

6. PLOS authors have the option to publish the peer review history of their article (what does this mean?). If published, this will include your full peer review and any attached files.

Reviewer #1: No

Reviewer #2: No

---

## [Author Response · Author response to Decision Letter 0]

9 Apr 2024

Dr Dariusz Kaczmarczyk Olsztyn, 18. 03. 2024

Department of Pond Fisheries

National Inland Fisheries Institute in Olsztyn

Editor of PLOS ONE Journal

Statement about Editor and Reviewer comments and suggestions

Dear Editor,

I write in response to review the paper with the Editorial Reference Number: PONE-D-24-01902. I would like to thank Editor and anonymous Reviewers for reviews this manuscript and for valuable suggestions. The manuscript was reorganized and ameliorated according to Reviewer comments.

The list of major changes in this revised version is given bellow. To make this answer clear and the changes easily identifiable in revised version all lines are given bellow are after they appear in clear version of this manuscript. 

A. The answers for a editor’s comments

A. According to editor’s suggestions a format of the manuscript was changed to fit it to PLOS ONE's style requirements

2. To comply with PLOS ONE submissions requirements, in your Methods section, please provide additional information regarding the experiments involving animals and ensure you have included details on (1) methods of sacrifice, (2) methods of anesthesia and/or analgesia, and (3) efforts to alleviate suffering, 3. In your Methods section, please provide additional information regarding the permits you obtained for the work. Please ensure you have included the full name of the authority that approved the field site access and, if no permits were required, a brief statement explaining why. 6. Please include your full ethics statement in the ‘Methods’ section of your manuscript file. In your statement, please include the full name of the IRB or ethics committee who approved or waived your study, as well as whether or not you obtained informed written or verbal consent. If consent was waived for your study, please include this information in your statement as well.

A. The Methods section was supplemented with information about investigation of fish. This information was included in the manuscript in lines 126-135. A details of permits given by Local Ethics Commission in Olsztyn were added in Methods section (lines 132-135). Moreover, we gave some additional information in Section Study area, translocations and fish sampling

4. We suggest you thoroughly copyedit your manuscript for language usage, spelling, and grammar. If you do not know anyone who can help you do this, you may wish to consider employing a professional scientific editing service. 

A. Copyediting was performed by an professional English language linguist. The statement was added as a supplemental file to the revision.

A. We thank for proposition of take advantage of American Journal Experts (AJE). The details of professional service that edited our manuscript are included as a supplementary file.

7. When completing the data availability statement of the submission form, you indicated that you will make your data available on acceptance. We strongly recommend all authors decide on a data sharing plan before acceptance, as the process can be lengthy and hold up publication timelines. Please note that, though access restrictions are acceptable now, your entire data will need to be made freely accessible if your manuscript is accepted for publication. This policy applies to all data except where public deposition would breach compliance with the protocol approved by your research ethics board. If you are unable to adhere to our open data policy, please kindly revise your statement to explain your reasoning and we will seek the editor's input on an exemption. Please be assured that, once you have provided your new statement, the assessment of your exemption will not hold up the peer review process

A. Following of editors’s recommendations we decided to deposit a results of our studies (as a pdf file in MSA format). Those data are available at: https://osf.io/62tby/?view_only=513c41b6d7e547c0a558b116a0a8374e

A. In the section Financial Support Statement we add a role of each sponsor in conducting of the study and preparation of this manuscript. 

A. Figure 1 is an original schematic map created specifically for the present manuscript by its authors using open-source data and Corel Draw X6 software. As the copyright owners, we allow the use of this map under the terms of the Creative Commons Attribution License 4.0.

A. I have used the PACE tool to improve the quality of figures that are submitted in this revision. 

B. The answers for a reviewer’s comments

1. (The PLOS Data policy requires authors to make all data underlying the findings described in their manuscript fully available without restriction, with rare exception (please refer to the Data Availability Statement in the manuscript PDF file). The data should be provided as part of the manuscript or its supporting information, or deposited to a public repository. For example, in addition to summary statistics, the data points behind means, medians and variance measures should be available. If there are restrictions on publicly sharing data—e.g. participant privacy or use of data from a third party—those must be specified.)

A. The genotyping data was submitted to OSF depository at address: https://osf.io/62tby/?view_only=513c41b6d7e547c0a558b116a0a8374e and are available. 

Reviewer#1

1. Line 122: Please revised Eupallasella percnurus to E. percnurus.

A. This fragment was changed according to reviewer’s suggestion and is now at Line 108. 

2. Line 248: Please change K 8-17 to K 8, 10, 11, 13, 15, 17.

A. This fragment was changed according to reviewer’s suggestion and is now at Line 288.

3. In line 240, why authors reported that K=11 was identified as the optimum value?

A. “K=11 was identified as the optimum value of this parameter because at this level the population structure, their relationship and admixture status were very close to field observations and results of our previous studies. “ This fragment is now at Line 259-261 in the revised version of this manuscript.

Reviewer#2

General comment. This manuscript was constructed around the questions of how a genetic variation in a group of populations established through a conservation program of this species differs between a group of decades-old populations, and what the perspectives are of using them in ongoing conservation programs. Although tracking individual differences between lake minnow populations is always interesting, this subject is the primary aim of this manuscript, likewise, the technical aspects of molecular method (null alleles presence) used and molecular markers selection (loci under selection). Following the reviewer’s suggestions, appropriate tests were performed, their details and results were given in the Methods section but they were reported briefly in the Results section and discussed shortly in the discussion section. 

The materials and methods are adequate for DNA extraction and microsatellite amplification. Still, the resulting database should allow for much more detailed analysis (unfortunately the raw data are not available in the submitted material, although the statement suggests that they will be available after publication.)

1. Before examining diversity and genetic structure, the presence of null alleles and loci under selection should have been investigated. The latter would also provide additional information for the objective.

A. Following reviewer’s suggestions we tested a probability of null alleles presence and loci under selection. In the revised version of this manuscript the details of null alleles presence test were given at lines 159-167 and identification of loci under selection at lines 184-188. The results of those test were reported in Results section at line 226-231.

2. In addition to calculating diversity indices, detecting private alleles and their frequencies in populations could provide important information.

A. Following reviewer’s suggestion we added those calculations to the manuscript text in the sections: Methods line 187-190, Results 264-267, 269-283, and discussion 375-376 and 378-399

3. If data on the original source population are available, allele distributions can provide additional information on the changes in diversity. It would be important to know the diversity of the source population since each of the new populations mostly showed higher diversity than the 13 old populations.

A. Unfortunately no data of original source population is available, because it was lost shortly after conservation program was started. 

4. Values of the G-W index less than 0.70 suggest that all of the new and old populations have experienced a bottlenecking or founder event.

A. Thank you for this comment, I have added this information to the Results section line 239-241.

5. Based on the genetic structure and diversity data, it appears that the translocations did not occur from a single source population in all cases. 

A. Those populations were established by used the progeny obtained during different periods of time and spawning. Although all spawners were taken from the same population, Its likely that some of differences between populations can be a consequence of using different sets of spawners in production of stocking material at given year. This topic was discussed in Discusion section (line 376-384)

6. The original lnPD values should be also presented for the Stucture analysis. 

A. The InPD values were calculated as an average for all iteration of Structure runs at given K were introduced to Figure 3. The methods of their calculation was mentioned in Methods section at line 182-183 and caption of Figure 3 at line 294. 

7. The delta K method indicated 13 clusters in the analysis, resulting in N03 and N05 being different from the other three new populations. 

A. Probably due the founder effect, some population specific alleles and genetic drift the populations N03 and N05 differs to other new populations. This was indicated in the Results section at line 264 – 267 results. A possible explanation of those differences was added to Discussion section at line 376-284. 

8. Moreover, N03 shows the origin of the BLD population and N05 is a unique cluster, based on Figure 3.

A. We checked for similarities between these populations using genetic distance estimation based on the fixation index (FST) and variation average allelic size (δμ2). The results of these calculations have not confirmed the similarity of the N03 and BLD populations. Both models show large and very large differences between these populations. Possibly this similarity is an artifact of STRUCTURE software.

9. This latter one could be the result of private alleles remaining from the original population which are absent from the other three due to the founder effect;

A. Indeed, this population had five private alleles that were not observed in any other populations from the new populations. This information was added to the Results section line 264-267 and was discussed in the Discussion section at line 376-384

 10. or the admixture with a local population that was not detected before the translocation. 

A. To the best of our current knowledge, all new populations are strictly isolated from other lake minnow populations. Thus, gene flow between new and other populations of this fish is not a factor that contributes to their genetic variation. This topic was introduced to the Discussion section at lines 392-394.

10. The results should be re-evaluated according to the findings from the Structure analysis.

A. Results section was supplemented by results of STRUCTURE analysis and other analysis. 

11. The diversity data could be correlated with the translocated individual number (founder population size) to estimate its effect.

A. Thank you, reviewer for this suggestion. Indeed a new population N04 is that one where the number of fish used to establish it was lowest. This population had the lowest heterozygosity and number of alleles. This information was added to the discussion (line 370-372) and this population was marked in results where values of genetic variation indicators are given at line 205, 233.

---

## [Decision Letter · Decision Letter 1]

9 May 2024

Genetic variability of the endangered fish lake minnow (Eupallasella percnurus) in populations newly established by translocation and those existing long term in Poland

PONE-D-24-01902R1

Dear Dr. Kaczmarczyk,

We’re pleased to inform you that your manuscript has been judged scientifically suitable for publication and will be formally accepted for publication once it meets all outstanding technical requirements.

Kind regards,

Tzen-Yuh Chiang

Academic Editor

PLOS ONE

Additional Editor Comments (optional):

Reviewers' comments:

Reviewer's Responses to Questions

**Comments to the Author**

1. If the authors have adequately addressed your comments raised in a previous round of review and you feel that this manuscript is now acceptable for publication, you may indicate that here to bypass the “Comments to the Author” section, enter your conflict of interest statement in the “Confidential to Editor” section, and submit your "Accept" recommendation.

Reviewer #1: All comments have been addressed

Reviewer #2: All comments have been addressed

2. Is the manuscript technically sound, and do the data support the conclusions?

Reviewer #1: Yes

Reviewer #2: (No Response)

3. Has the statistical analysis been performed appropriately and rigorously? 

Reviewer #1: Yes

Reviewer #2: (No Response)

4. Have the authors made all data underlying the findings in their manuscript fully available?

Reviewer #1: Yes

Reviewer #2: (No Response)

5. Is the manuscript presented in an intelligible fashion and written in standard English?

Reviewer #1: Yes

Reviewer #2: (No Response)

6. Review Comments to the Author

Reviewer #1: The authors have addressed all comments. The manuscript is suitable to be accepted for publication.

Reviewer #2: (No Response)

7. PLOS authors have the option to publish the peer review history of their article (what does this mean?). If published, this will include your full peer review and any attached files.

Reviewer #1: No

Reviewer #2: No

---

## [Editor Report · Acceptance letter]

23 May 2024

PONE-D-24-01902R1 

PLOS ONE

Dear Dr. Kaczmarczyk, 

I'm pleased to inform you that your manuscript has been deemed suitable for publication in PLOS ONE. Congratulations! Your manuscript is now being handed over to our production team.

Kind regards, 

on behalf of

Dr. Tzen-Yuh Chiang 

Academic Editor

PLOS ONE